# VaBTFER: An Effective Variant Binary Transformer for Facial Expression Recognition

**DOI:** 10.3390/s24010147

**Published:** 2023-12-27

**Authors:** Lei Shen, Xing Jin

**Affiliations:** College of Information Science and Technology, Nanjing Forestry University, NanJing 100190, China; leis@njfu.edu.cn

**Keywords:** facial expression recognition, spatial-channel feature relevance Transformer, lightweight variant Transformer, binary quantization mechanism, multilayer channel reduction self-attention, dynamic learnable information extraction

## Abstract

Existing Transformer-based models have achieved impressive success in facial expression recognition (FER) by modeling the long-range relationships among facial muscle movements. However, the size of pure Transformer-based models tends to be in the million-parameter level, which poses a challenge for deploying these models. Moreover, the lack of inductive bias in Transformer usually leads to the difficulty of training from scratch on limited FER datasets. To address these problems, we propose an effective and lightweight variant Transformer for FER called VaTFER. In VaTFER, we firstly construct action unit (AU) tokens by utilizing action unit-based regions and their histogram of oriented gradient (HOG) features. Then, we present a novel spatial-channel feature relevance Transformer (SCFRT) module, which incorporates multilayer channel reduction self-attention (MLCRSA) and a dynamic learnable information extraction (DLIE) mechanism. MLCRSA is utilized to model long-range dependencies among all tokens and decrease the number of parameters. DLIE’s goal is to alleviate the lack of inductive bias and improve the learning ability of the model. Furthermore, we use an excitation module to replace the vanilla multilayer perception (MLP) for accurate prediction. To further reduce computing and memory resources, we introduce a binary quantization mechanism, formulating a novel lightweight Transformer model called variant binary Transformer for FER (VaBTFER). We conduct extensive experiments on several commonly used facial expression datasets, and the results attest to the effectiveness of our methods.

## 1. Introduction

Face expression can intuitively reflect the innermost true thoughts of humans. Numerous research studies have been conducted on autoexpression analysis because of its important role in various research fields such as human–computer interaction, emotion analysis, and healthcare [1,2,3]. It has been a hot problem to develop highly effective methods for feature extraction including DNN methods [4,5] and traditional machine learning methods [6,7,8,9]. Existing effective FER methods also mainly include two categories: those that rely on traditionally handcrafted feature extraction(e.g., local binary patterns (LBP) [10], non-negative matrix factorization (NMF) [11], and sparse learning [12]), and those that utilize deep neural networks (DNNs) (e.g., convolutional neural networks (CNNs) and recurrent neural networks (RNNs)). DNN-based methods for the second category have demonstrated remarkable performance, owing to their robust feature extraction and pattern learning capabilities. However, according to the Facial Action Coding System (FACS) [13,14,15], various facial expressions can be produced by the specific movements of facial muscles [16]. Traditional convolution operations are limited to fixed receptive fields, which results in an inability to model the long-range relationships between facial muscle movements [17]. On the contrary, RNNs excel in modeling long-range dependencies in sequence data by leveraging their cyclic structures, leading to superior performance in comparison. Nonetheless, they are susceptible to encountering gradient vanishing or explosion issues during backpropagation. Moreover, RNNs are incapable of parallelization operations and also cannot harness the exceptional parallelization capabilities of graphic processing units (GPUs) while running recursively. It is a promising topic to model muscle movements through popular DNNs [18,19].

Recently, the Transformer [20] has made its debut in natural language processing (NLP) [21,22]. It exploits the self-attention mechanism (SAM) to model the long-range dependencies based on parallelization processing. There has been a surge in the application of the Transformer architecture in computer vision (CV) tasks [23,24] since Vision Transformer (ViT) [25] has been introduced, particularly in FER where a family of excellent Transformer-based models has been designed by modeling complex facial muscle movements and shown more promising results than CNNs [26,27]. For example, Zeng et al. [28] presented Face2Exp for FER, which comprised a base network and an adaptation network. In [29], Kim et al. proposed utilizing multimodal information and a Swin Transformer (ST) to enhance the performance of the FER model. Unfortunately, deriving from the use of the pure Transformer encoder, the pioneering models are inevitably equipped with a large number of parameters, leading to many difficulties in model deployment. Compared with CNNs, the Transformer itself also lacks inductive bias, and in it, the prior information must be learned through a large quantity of data to achieve better performance [25]. In addition, the multilayer perception (MLP) utilized in the Transformer encoder cannot effectively strengthen the relations among all tokens, as they merely perform a simple dimensional stretch on the tokens themselves. Therefore, to achieve superior results with limited FER datasets, it is imperative to design a variant Transformer.

It is widely acknowledged that running DNNs efficiently on low-power devices is a daunting task, and considerable research efforts are focused on boosting the speed of DNNs [30,31,32]. Among them, a representative method is binarized networks [32]. It utilizes binary weights and activation functions to significantly reduce the memory size and replace many numeric computations during the network’s forward process, which aids in deploying DNNs in resource-constrained devices. Binarized networks have sparked many studies on binary deep models for fast FER tasks. For example, Ferraz et al. [33] proposed a local binary CNN which reduced the number of parameters in the model and performed well during training. Kumawat et al. [34] proposed a novel local binary volume CNN, which, however, ignored the relationships between facial expressions and muscle movements. Currently, it is of utmost urgency to design a lightweight and efficient binary variant Transformer for FER.

To alleviate the above problems, in this paper, we propose a novel lightweight variant binary Transformer for FER (VaBTFER) with an effectiveness guarantee, which is based on ViT [25] and binary operations. Instead of simply dividing an image into patches that include much useless information, we adopt AU-based tokens, which are represented by histogram of oriented gradient (HOG) features [35], to replace the original image patch tokens. The purpose of doing this is to decrease any unwanted noise in the background and limit its input to the network. Then, an efficient spatial-channel feature relevance Transformer (SCFRT) module is designed to solve the problem of muscle movement relationship establishment, large parameter quantity, lack of inductive bias, and MLP failing to make tokens connect with each other. For the final expression prediction, the excitation module is also reused to enforce the interaction between channels. Finally, we introduce binary operations to enable the proposed VaTFER model to reduce floating-point operations and memory access with low-scale parameters, allowing for easy training on mobile devices [36]. The leading model is called variant binary Transformer for FER (VaBTFER). In the following, we summarize the main contributions of this paper:

(1) We propose a novel efficient Transformer framework for FER named VaTFER. The proposed model can obtain a higher recognition accuracy with fewer model parameters. Moreover, the proposed model can be effectively trained on small facial expression datasets.

(2) We propose an efficient spatial-channel feature relevance Transformer module which contains two modules (MLCRSA and DLIE) to handle more parameters and missing inductive bias in the traditional Transformer encoder. In MLCRSA, we adopt the 1D convolutional layer to get a decreasing-dimension query, key, and value. In addition, we only use single-head self-attention (SSA) rather than MSA. The DLIE module introduces an inductive bias and also acts as an attention mechanism to better learn the global connections among all AU-tokens.

(3) We extend VaTFER with the binary quantization mechanism to construct VaBTFER for FER. To evaluate the proposed models, we conduct extensive experiments on several FER datasets, including RaFD, CK+, and OULU-CASIA. The experimental results indicate that our models can be guaranteed to obtain comparable performance but requires much fewer model parameters as compared with other DNNs.

## 2. Related Works

To make the paper easier for the reader to follow, we briefly review some related works in this section from three aspects, i.e., FER based on DNNs, Transformer for FER, and binarized neural networks.

### 2.1. FER Based on Traditional Deep Neural Networks

Research on DNNs for FER, which exploits static face images and videos, has been promising. There have been a lot of scholars studying in this field. For instance, Niu et al. [37] proposed a new selective element and two-order vectorization (SE-TOV) mechanism to weaken the loss of fine clues about facial expression induced by the max-pooling or average-pooling operations. In the mechanism, the multiscale information could also be learned well without pooling operations. To enhance the performance and expedite the training and calculation of DNNs in FER, Huang et al. [38] proposed a multiplication framework that replaced the original convolutional operations with efficient element-wise multiplications. Moreover, this framework facilitated the retention of informative feature components through a representation scheme in the Fourier domain. Zhang et al. [39] proposed an FER model, which sought an expression-related area accurately and improved recognition accuracy in the visual geometry group (VGG) framework by incorporating three modules: the expression feature extractor (EFE), the expression mask refiner (EMR) and the expression pattern-map generator (EPMG). To better utilize the time sequence information, Meng et al. [40] developed a frame attention network (FAN) that automatically identified keyframes. In the network, the acquired attention weights were used for aggregating the video features to attain a discriminative video representation. Lee et al. [41] presented context-aware emotion recognition networks (CAERNets) to extract facial emotion and context-aware information by using two-stream encoding networks. All these methods are based on traditional CNNs and RNNs. Through local connections and weight-sharing mechanisms, a traditional CNN has the strong ability to extract local features and its training efficiency is high. However, the receptive field of convolution operation is limited, and it is impossible to model the long-distance dependency of facial expression features. RNNs [42] have a loop-connected structure and are suitable for processing sequence data. They can capture the evolution process of sequence information and have a relatively low training efficiency, making them difficult to parallelize.

### 2.2. Transformer for FER

The Transformer [20] was firstly proposed by Vaswani et al. for handling tokens in parallel and it became the general backbone in many NLP tasks. In 2020, Dosovitskiy et al. [25] proposed a new ViT that trained the model on image classification tasks (e.g., FER) and demonstrated promising results. Since then, numerous Transformer-based models designed for vision-related tasks have garnered significant attention from researchers [43,44,45,46]. These Transformer-based models split an image into tokens. They are subsequently fed into the Transformer to leverage its impressive self-attention mechanism to effectively model the long-range relationships among the tokens, which are key to improving the performance in FER. It is straightforward to consider utilizing the Transformer model to establish a correlation between facial muscle movements in face expression images. Li et al. [43] developed a pure Transformer-based mask vision Transformer (MVT) model, which involved two modules: a Transformer-based mask generation network and a dynamic relabeling module. The first module was used to generate a mask that could filter out complex backgrounds and occlusion of face images. The second module was designed to rectify incorrect labels in FER datasets. Zhang et al. [44] used a unified Transformer-based multimodal model for both AU detection and final expression recognition, which extracted multimodal information from static images and neighboring frame images. Xue et al. [46] proposed a novel model named TransFER, which mainly comprised multiattention dropping (MAD), ViT-FER, and multihead self-attention dropping (MSAD). MAD was used to locate discriminative patches and diverse local patches. ViT-FER was proposed to utilize a multi-head self-attention (MSA) mechanism to model patch relations. MSAD was used to guide the dismissal of similar relations between different local patches that were acquired by MSA. Zheng et al. [45] designed a two-stream pyramid cross-fusion Transformer (POSTER) network. It adopted a Transformer-based cross-fusion paradigm to calibrate the model focus on salient regions and a pyramid architecture that improved scale invariance. Although all these Transformer-based models have achieved high performance in FER, they require a large number of parameters. Moreover, these models are not trained from scratch on limited FER datasets but use pretrained weights on large datasets directly. How to develop a lightweight Transformer-based model by using a brilliant SAM module for FER is a research hotspot.

### 2.3. Binarized Neural Networks

DNNs have shown impressive performance in many fields; however, these achievements depend on a sophisticated network design and huge computational resources [47]. To take full advantage of DNNs on a mobile device, copious model compression techniques, i.e., pruning [48,49], low-rank factorization [50], quantization [51], and knowledge distillation [31] are used, where network quantization technology has become an important research direction in model compression. Binarization neural networks (BNNs) focus on quantizing weights and activation values to +1 and −1, and they replace high-precision 32-bit float parameters [36], thus alleviating the consumption of float arithmetic and internal memory. In 2016, Hubara et al. [52] first presented BNNs, which demonstrated comparable accuracy in smaller datasets. Nevertheless, when implementing the binary operations on larger datasets, their accuracy underwent a decline.

A lot of work is devoted to narrowing the gap between binary weights and real-valued weights. Rastegari et al. [36] proposed XNOR-Net, which introduced efficient approximations on standard CNN and linear layers to reduce the quantization errors by adding a fractional amount of real-valued part (a real-valued scaling factor) in the binary forward progress. Tu et al. [47] presented a simple but efficient mechanism named AdaBin to get the optimal binary weights, which used adaptive binary sets $\left\{b1,b2\right\}\left(b1,b2\in R\right)$ instead of fixed binary sets $\left\{+1,-1\right\}$ to fit different distributions and improve the representation ability. Liu et al. [53] developed ReActNet in which an elaborate real-valued network with parameter-free shortcuts was modified and binarized to narrow the gap between real-valued networks and binarized-valued networks enormously. Currently, there are relatively few binary networks used to solve the FER problems. To enhance the effectiveness of FER networks on mobile devices, we suggest introducing a lightweight binary network for emotion analysis.

## 3. The Proposed VaBTFER Model

In this section, we describe our VaBTFER model in detail. Firstly, we provide an overview of the overall framework of the proposed VaTFER, then present the SCFRT and excitation modules included in VaTFER. Finally, we design a binarization network in VaBTFER by incorporating a binarization strategy into the proposed VaTFER model.

### 3.1. Overview

The core of our work includes the following aspects: (1) compromising between model accuracy and lightweight design, (2) making use of the Transformer architecture to model long-range dependencies about all AU tokens and alleviating the inductive bias of the Transformer, (3) introducing a high-performance binary network strategy. To meet the above requirements, we design a novel deep model named VaTFER which includes two modules, i.e., SCFRT and excitation modules. The proposed VaTFER is shown in Figure 1. At last, we extend VaTFER with binarization operations to construct the VaBTFER model.

According to the research on FACS, facial expression is related to muscle movements. Therefore, it is necessary to model long-range relationships among all the movements. Our VaTFER, which is based on ViT, is proposed to acquire such relationships. We crop 20 AU-based regions of interest (ROIs) in a facial expression image, then an HOG is adopted to perform feature extraction for these ROIs, aiming to replace the original patch embedding in ViT [25]. In ViT, the features after patch embedding are called tokens, as in the NLP application. Therefore, the AU features after the HOG feature extraction are called AU tokens. These AU-based tokens can greatly reduce the effect of background and lighting noise in the facial expression image; thus, they can effectively help the proposed VaTFER model converge faster and simultaneously achieve better performance in the absence of a large quantity of training data.

As in ViT, a learnable class token is employed for the final FER classification in the proposed model. After adding the class token, all processed tokens including AU-based tokens and class token are fed into our SCFRT. In SCFRT, we first apply our MLCRSA to decrease the number of parameters, learn the tokens’ channelwise information and learn processed tokens’ long-range dependencies that are realized by a 1D convolution and single-head self-attention. After applying the MLCRSA module, we utilize our DLIE module to introduce the inductive bias and facilitate the connection of AU-based tokens with one another. In brief, we utilize a CNN to introduce the inductive bias. The AU tokens are processed by the CNN and then combined with the original AU-based tokens to guarantee the derivation of the spatial bias. Afterward, average pooling is utilized to obtain a comprehensive weight that considers all AU tokens in terms of space. The obtained weights are used to enhance the channelwise connections through the excitation module. Then, it is utilized for multiplication with the class token. By doing so, the class token can acquire both the spatial and channelwise information from the AU tokens, which is not achievable with the MLP in the original Transformer. The class token is used for the final prediction. To further enhance channelwise relationships of class token in the final prediction, the universal excitation module is reintroduced in our model instead of using basic an MLP for the prediction. To reduce floating-point operations and memory access, we further design a VaBTFER model, which introduces binary quantitative operations to the VaTFER model, in which we approximate the SCFRTs to the formulated binary networks by adding a scaling factor and modifying the backward gradients. The involved modules are listed in the following subsections.

### 3.2. Spatial-Channel Feature Relevance Transformer

Certain issues arise when using transformers for FER tasks. One such issue is the requirement for a large number of parameters and extensive training data [25]. Additionally, transformers lack an inductive bias, which poses a challenge when training models from scratch with limited FER datasets. Moreover, the original FFN cannot strengthen the connection among all tokens since it scales the dimension simply. Thus, we propose a new SCFRT model to mitigate these problems, which consists of two submodules: MLCRSA and DLIE. 

#### 3.2.1. Multilayer Channel Reduction Self-Attention

For a given facial expression image $I_{in}\in R^{H\times W\times C}$, where *H* and *W* denote the height and width of a facial expression image, and *C* the channel of the image, we utilize 20 AU-based tokens and one class token mentioned above to replace the original patch embedding in ViT [25]. Then, these preprocessed tokens become the input to our MLCRSA. We introduce one layer of our MLCRSA called channel reduction self-attention. To be specific, the layer normalization is firstly applied to perform normalization. Then, the output of the layer normalization is defined as $X\in R^{N\times L}$, where *N* is 21, that is, the number of tokens, and *L* is the length of each of tokens. Then, the 1D convolution operation is adopted to get the query *Q*, the key *K*, and the value *V* that are defined the same as in ViT [25]. The acquirement of Q, K, and V can be expressed with the following formulations:(1)$$\begin{array}{c} Q=XW_{Q},K=XW_{K},V=XW_{V}, \end{array}$$
where $W_{Q},W_{K}$, and $W_{V}$ are the learnable 1D convolution weights, respectively, and the output *Q*, *K*, and *V* are in $R^{N\times d}$, where *d* denotes the reduced dimension. The 1D convolution operation is used to model the channel features and decrease the dimensions of tokens *X* in parallel. Then, single-head self-attention is used to capture the long-range connections among all these tokens which is shown with the following equation:(2)$$\begin{array}{c} Atten_{value}=Softmax\left(\frac{QK^{T}}{\sqrt{d}}\right)V. \end{array}$$
where Softmax(·) is a function that is used to map real values between (0,1) and $Atten_{value}$ has the same shape as *Q*, *K*, and *V*, i.e., $R^{N\times d}$. We stack each layer including the layer normalization operation, Equations (Equation 1) and (Equation 2), and this is our proposed MLCRSA. MLCRSA can further reduce the parameters by using 1D convolutional layers to reduce the dimension of tokens (from L to d) in each layer of MLCRSA and model long-range dependencies using single-head self-attention among all tokens. 

#### 3.2.2. Dynamic Learnable Information Extraction

After the self-attention mechanism is applied, the attention value is promptly scaled by the MLP in ViT [25]. The original MLP in ViT is composed of two linear layers and an activation function, which can be described as:(3)$$\begin{array}{c} X_{F}=Linear\left(gelu\left(Linear\left(Atten_{value}\right)\right)\right), \end{array}$$
where $X_{F}$ denotes the output of our $Atten_{value}$ after two fully connected layers that are described with Linear(·) in Equation (Equation 3), and gelu(·) refers to a smooth nonlinear activation function called a Gaussian error linear unit (GELU). It is clear that the MLP just utilizes fully connected layers to reduce and increase the dimensions of tokens. Therefore, the MLP only focuses on the tokens’ channel dimension that lacks communication between tokens. To address this issue, we propose an innovative DLIE module. The DLIE module firstly introduces inductive bias and makes AU-based tokens connect each other in receptive fields by reshaping all the AU-based tokens excluding the class token, and then incorporating the spatial inductive bias which is fulfilled by a 2D convolution. The total procedure is shown as follows:(4)$$\begin{array}{c} X_{class}={Spl}_{c}\left(Atten_{value}\right), \end{array}$$
(5)$$\begin{array}{c} X_{AU}={Spl}_{a}\left(Atten_{value}\right), \end{array}$$
(6)$$\begin{array}{c} X_{r}=Reshape\left(X_{AU}\right), \end{array}$$
(7)$$\begin{array}{c} X_{adj}=Conv(gelu\left(Conv\left(gelu\left(Conv\left(X_{r}\right)\right)\right)\right), \end{array}$$

As shown in Equations (Equation 4) and (Equation 5), we split the AU tokens and class token from $Atten_{value}$ using functions $Spl_{c}\left(\cdot \right)$ and $Spl_{a}\left(\cdot \right)$, which are represented by $X_{AU}$ and $X_{class}$, respectively. In Equation (Equation 6), Reshape(·) is used to represent the function that rearranges our AU-based tokens in the shape of H×W×C, which enables us to make use of a 2D convolution to incorporate a spatial inductive bias and model the neighboring relations of AU-based tokens. $X_{r}$ is denoted as the output of the reshaped tokens. As shown in Equation (Equation 7), the 2D convolutional layers are described as Conv(·), and the CNN and GELU activation functions are applied to capture the adjacent features effectively. We use $X_{adj}$ to represent these processed tokens and fuse them with $X_{r}$, and the AU tokens are finally expressed with $X_{s}=X_{adj}+X_{r}$. With the spatial inductive bias brought by the convolutional operation, the AU tokens in the DLIE module can interact with each other effectively.

In addition, the DLIE module can also help the class token $X_{class}$ learn information about all the AU-based tokens $X_{s}$ that is beneficial for final prediction. Firstly, we learn a weight that represents the information about all $X_{s}$. To be concrete, the processed tokens $X_{s}$ are fed into average-pooling layers to get a weight which is the same as that of the class token in terms of dimension. This is depicted in the following equation:(8)$$\begin{array}{c} X_{weight}=Averpool\left(X_{s}\right), \end{array}$$

Then, the weight is remodeled on the channel dimension by compressing and enlarging the size of the channel dimension that is the same as that of the excitation block, which is clearly depicted in Section 3.3. After that, the learned weight after the excitation block is defined as $X_{co}$, and $X_{co}$ is multiplied with the class tokens $X_{class}$ in Equation (Equation 4). We utilize $X_{cls}$ to represent the resulted values. To this end, the formulated problem can be written as follows:(9)$$\begin{array}{c} X_{cls}=X_{class}⊙X_{co}. \end{array}$$
where $X_{cls}$ can be viewed as a weighted version of $X_{class}$, which plays a crucial role in aggregating the global information from all the AU-based tokens, across both spatial and channel perspectives. The operation mentioned above is designed to compensate for the missing performance from the utilization of the single-head self-attention mechanism in MLCRSA. This improvement significantly enhances the model’s learning ability. The whole DLIE module can establish connections between the AU-based tokens and model the neighboring tokens within the receptive fields by introducing a spatial bias. Additionally, it enhances the class token’s capacity to obtain spatial and channelwise information from AU-based tokens. Combining DLIE with MLCRSA formulates our SCFRT module. This hybrid module has the potential to narrow the performance gap with standard CNNs when training our model from scratch on limited-scale FER datasets [54].

### 3.3. Excitation Module for Final Prediction

After SCFRT, we can use a simple MLP and a fully connected layer for the final facial expression prediction, as in ViT [25]. However, the sole use of an MLP may not effectively model the channelwise information of $X_{cls}$ across channels, as shown in Equation (Equation 3). As we all know, the excitation module acts as a generic performance-enhancing module by explicitly modeling channelwise information among the channels, and it produces significant performance improvements at minimal additional computational cost [55,56]. As a result, we use the excitation module to replace the MLP, which can be formulated as follows:(10)$$\begin{array}{c} EXC_{cls}=Sigm(Linear_{\gamma }\left(relu\left(Linear_{\gamma /n}\left(X_{cls}\right)\right)\right), \end{array}$$
where Sigm(·) and relu(·) denote the sigmoid and rectified linear unit (RELU) activation functions, respectively, which improve the generalization of the model. $Linear_{\gamma /n}\left(\cdot \right)$ refers to a dimensionality-reduction fully connected layer with reduction ratio *n*, and $Linear_{\gamma }\left(\cdot \right)$ means using a fully connected layer to restore the token dimension to the original γ. From Equation (Equation 10), $X_{cls}$ is reduced by reduction ratio *n* and is then restored by fully connected layers. Through all these operations, the channelwise information $EXC_{cls}$ is captured by $X_{cls}$ using a simple pointwise(⊙) operation in Equation (Equation 11).
(11)$$\begin{array}{c} X_{final}=X_{cls}⊙{EXC}_{cls}. \end{array}$$
where $X_{final}$ is the output of the last excitation module, and it is used for the final facial expression prediction using a linear layer.

### 3.4. Binary Strategy

In the previous section, we present a novel VaTFER. To better deploy the proposed VaTFER on mobile devices with limited memory and computation resources, the binary operation is introduced to compress the size of the model, reduce the memory, and accelerate the inference speed in the network forward propagation. However, the simple and brutal binarization strategy of using only binarizing weights leads to a lower recognition accuracy. Based on [32], an accurate and speedy binarization is designed which improves the accuracy of FER tasks compared to the simple binarization strategy mentioned above. We introduce our binarization strategy in the following subsections.

#### 3.4.1. Binarizing Weight and Activation Value in Forward Propagation

In this subsection, we introduce our binary weight and activation in the forward progress. The binarization function is firstly introduced:(12)$$\begin{array}{c} R^{b}=sign\left(R\right)=\left\{\begin{array}{cccc} -1 & \; & if\;R⩽0, & \\ +1 & \; & otherwise, \end{array}\right. \end{array}$$
where $R^{b}$ denotes the binarization value (weight or activation), and *R* denotes the 32-bit float real value. Moreover, in Equation (Equation 12), function sign(·) is used to perform binarization operations according to the range of real value *R*. For a convolutional or linear layer <X,W,*>, X denotes the input tensor of the convolutional or linear layer, W is the weight of the convolutional or linear layer, and * denotes the arithmetic symbols of the convolutional or linear layer. According to [32], the values in W are computed as either +1 or −1 by Equation (Equation 12). Many studies have shown that this kind of simple binary function does not work well [32]. To tackle this issue, we devoted ourselves to finding an approximate solution to W. To be specific, we learned a new binarized weight **B** obtained by using the sign(·) function on W and a scaling factor β and used their product to approximate W. This can be elegantly expressed with the following approximate linear function:(13)$$\begin{array}{c} W\approx \beta B, \end{array}$$
Note that W and B are vectors in $R^{n}$ and n is the number of weights W. We can apply the following least squares loss function to learn the function:(14)$$\begin{array}{c} \left(\beta^{*},B^{*}\right)=\underset{\beta ,B}{argmin}J\left(\beta ,B\right), \end{array}$$
where $J\left(\beta ,B\right)={\left\|W-\beta B\right\|}^{2}$. By expanding function J(β,B), we have
(15)$$\begin{array}{c} J\left(\beta ,B\right)=\beta^{2}B^{T}B-2\beta W^{T}B+W^{T}W. \end{array}$$

By minimizing Equation (Equation 15), we can conclude the optimal $\beta^{*}$ is as follows:(16)$$\begin{array}{c} \beta^{*}=\frac{W^{T}\mathrm{sign}(W)}{n}=\frac{\sum |W_{i}|}{n}=\frac{1}{n}{\left\|W\right\|}_{l_{1}}, \end{array}$$
where ${\left\|\cdot \right\|}_{l_{1}}$ denotes the $l_{1}$ norm. Equation (Equation 16) indicates that the optimal $\beta^{*}$ can be determined by calculating the average of the absolute values present in W. For activation value A that consists of vectors in $R^{n}$ and n is the number of activation values A, we binarize A simply using sign(·) so that the binarized activation value A can be presented in the form of $A^{b}=sign\left(A\right)$.

#### 3.4.2. Gradients’ Computation and Optimization in Backward Propagation

Our binarized weights are defined as $W^{b}$, and $W^{b}=\beta B=\beta sign\left(W\right)$ according to Section 3.4.2. In backward propagation, we need to update our real-valued W using gradient descent. Then, the gradient of W needs to be computed. As we all know, the gradients of the sign function are considered zero at every point that can be formulated as $\frac{\partial sign(W)}{\partial W}=0$. Therefore, it is difficult to compute the gradient of our weights W. We directly use the autogradient of binarized weights $W^{b}$ to replace the gradient of our weights. This is formulated as follows:(17)$$\begin{array}{c} g_{W}=g_{W^{b}}=\frac{\partial C}{\partial W^{b}}. \end{array}$$
where $g_{W}$ and $g_{W^{b}}$ denote the gradients of W and $W^{b}$, respectively, and *C* refers to our loss function.

Our binarized activation values are defined as $A^{b}=sign\left(A\right)$. We adopt a different strategy to compute the gradient about our activation value A. Bengio et al. [57] proposed the straight-through estimator (STE). The STE was proved to be accurate and speedy during training. The STE sets the autogradients to zero like dropout [58] when the autogradients’ absolute values are less than one, instead of randomly setting them to zero. Therefore, our final gradients of binary activation values A are computed using the STE as can be seen in Equation (Equation 18)
(18)$$\begin{array}{c} g_{A}=g_{A^{b}}1_{|A|⩽1}=\left\{\begin{array}{cccc} \frac{\partial C}{\partial A^{b}} & \; & if\;|A|⩽1, & \\ 0 & \; & otherwise, \end{array}\right. \end{array}$$
where $g_{A}$ is the gradient of A, and $g_{A^{b}}$ denotes the gradient of binarized activation $A^{b}$. $1_{|A|⩽1}$ means 1 when the absolute value of the real-valued activation A is less than or equal to 1, and 0 otherwise. In summary, $g_{A}$ is computed using $\frac{\partial C}{\partial A^{b}}$ if the absolute value of A is less than 1, otherwise the gradient $g_{A}$ is set to zero. Last but not least, the optimizer of our model is changed to the Adam optimizer rather than the original SGD [47].

## 4. Experiments

To evaluate the effectiveness of the proposed model, we conducted experiments on several FER datasets. Experimental details included the preprocessing of datasets and the comparison of our unbinarized model VaTFER with state-of-art DNNs regarding both their accuracy and model size. In addition, we estimated the effect of different modules in our proposed model and chose the best architecture for our final VaTFER. Finally, the results of the proposed VaTFER with binary operations are also displayed.

### 4.1. Datasets and Preprocessing

In this subsection, three FER datasets (i.e., Radboud Faces Database (RaFD) [59], Extended Cohn-Kanade (CK+) [60], and Oulu-CASIA [61]) were adopted in the experiments. These datasets were collected in a laboratory-controlled environment. However, the Oulu-CASIA dataset poses a greater challenge compared to the first two datasets due to its intricate background and diverse types of occlusions. Some samples for the three datasets are shown in Figure 2.

RaFD is composed of 8040 (67×5×3×8) facial images of 67 subjects in five different postures, three eye-gaze directions and eight face expressions. In this paper, we selected the 1474 frontal facial images (67×7×3) with the seven basic expressions (anger, disgust, fear, happiness, neutral, sadness, and surprise). CK+ is a widely used FER dataset and it consists of 593 video sequences captured by 123 subjects, and each subject’s facial expression changes from a neutral expression to different spontaneous expressions. However, only 327 video sequences had corresponding expression labels, including anger, disgust, neutral, happiness, sadness, and surprise. We used these 327 video sequences. In our experiments, only the last three frames of the video sequences were used as these frames were labeled with a corresponding video label. Therefore, our CK+ dataset consisted of 981 (327×3) images with six basic expressions. The last presented dataset, Oulu-CASIA, is similar to CK+ and also consists of video frame sequences. Oulu-CASIA includes 80 subjects and six basic expressions. It was collected by two imaging systems (near infrared and visible) under three different lighting conditions (dark, strong, and weak). In this experiment, we also chose the last three frames of each video under natural lightning conditions. As a result, our whole Oulu-CASIA dataset was composed of 1440 (80×6×3) facial expression images. The data preprocessing takes an important role in the FER task. In this experiment, we handled these selected images with the same preprocessing. We firstly used Dlib C++ to align the face region of each image in the three datasets. The face portion of each image was cropped and resized to 128×128. Then, Dlib C++ was used again to acquire facial landmarks, and these acquired landmarks were computed to get the AU regions. In the end, we handled the AUs using HOG features. We took the final preprocessed datasets as inputs for our proposed model. The accuracy of our model was determined through a fivefold cross-validation on RaFD and tenfold cross-validation on CK+ and Oulu-CASIA.

### 4.2. Experimental Results

We compared our unbinarized VaTFER with some existing large-scale and lightweight DNN models on RaFD, CK+, and Oulu-CASIA in terms of accuracy and parameters. The detailed results are shown in Table 1, Table 2 and Table 3.

#### 4.2.1. Results on RaFD

We conducted experiments on a fair basis by utilizing popular models such as ViT [62], MSC-ViT [62], SqueezExpNet-ABR [63], EDNN [64], PeleeNet [65], ResMoNET [66], L3Net [67], DDRGCN [68], and MicroExpNet [69]. These models are widely used on the RaFD dataset, which allows for a comprehensive and accurate comparison of our findings. As is shown in Table 1, our VaTFER achieves the second highest accuracy of 97.04% with 50K parameters, which are the least compared to other DNNs models. ViT obtains only a 54.34% accuracy using the largest number of parameters 86,000 K, and this confirms that the excellent ViT cannot achieve good results when training from scratch on limited datasets. The next Transformer architecture model, MSC-ViT, reaches the highest accuracy without pretrained weights by introducing a convolution operation in ViT that is similar to our VaTFER; however, the number of parameters of MSC-ViT is still huge.

We also compared our VaTFER with traditional CNNs that have a large number of parameters, such as SqueezExpNet-ABR, EDNN, PeleeNet, and ResMoNET. As seen in the table, the accuracies obtained by SqueezExpNet-ABR and EDNN were 96.50% and 95.00%, respectively, which was lower than that of our VaTFER. In addition, these two models required several times more parameters than our VaTFER. Both PeleeNet and ResMoNET needed fewer parameters than EDNN and ResMoNET, but their accuracies experienced an 11.00% and 5.00% decrease, respectively, compared with EDNN and ResMoNET. In the following, we also show some lightweight CNNs (i.e., L3Net and MicroExpNet) and graph convolutional models (i.e., DDRGCN). Compared with VaTFER, L3Net and DDRGCN showed inferior recognition accuracy and used twice as many parameters. MicroExpNet had a similar number of parameters to our model, but the recognition accuracy was much worse. In total, the experimental results convinced us that VaTFER could surpass other models in terms of recognition accuracy and required fewer parameters.

#### 4.2.2. Results on CK+

Table 2 shows the results of different models on CK+ dataset. As shown, our comparisons focused on several widely used DNNs models for FER on CK+, including Graph + Tran [70], ESTLNet [71], DMSRL-VF [72], ATSFDCNN-CONCAT [38], DSAN-VGG-GENDER [73] and DSAN-RES-GENDER [73], ST-BLNwo/MCD [74], AGCN [75], L3Net [67], DDRGCN [68], and MicroExpNet [69]. From the table, it can be observed that VaTFER achieved an accuracy of 97.49%, which was relatively high among all the models, but it required the least number of parameters. The Transformer network ESTLNet obtained the highest accuracy rate, which, however, was very similar to that of VaTFER. Another Transformer network, Graph+Tran, achieved the second highest recognition accuracy in all listed networks, which revealed the superiority of the Transformer network. Both DSAN-VGG-GENDER and DSAN-RES-GENDER employ different demographic-based attention knowledge. These two models take VGG and Resnet50 as the backbone and exploit extra gender labels. They required several hundred times more parameters than VaTFER to obtain a similar accuracy. DMSRL-VF uses VGG16 with pretrained weights as a feature extraction backbone. It only had a 91.26% recognition accuracy, far lower than VaTFER, but the number of parameters was much higher than VaTFER. The last larger-scale model, ATSFDCNN-CONCAT, achieves the second highest accuracy using 42,000 K parameters. Also, we can see that the lightweight ST-BLNwo/MCD and AGCN obtained 93.19% and 94.18% recognition accuracies but used approximately three times the number of parameters as VaTFER. The accuracy of L3Net and DDRGCN, which needed 102K parameters, was still nearly three points lower than that of our VaTFER model. While MicroExpNet utilized a comparable number of parameters as our model, its recognition accuracy was significantly lower.

#### 4.2.3. Results on Oulu-CASIA

Oulu-CASIA is more challenging than RaFD and CK+ because of its low image quality. For example, it includes complicated backgrounds and diverse types of occlusion. The models mentioned in the above subsection were applied to that dataset. The results are listed in Table 3. From the table, we can see some main points. For example, DMSRL-VF only acquired a 63.28% recognition accuracy with most model parameters. ESTLNet and Graph + Tran achieved the two highest accuracies of 89.38% and 89.31% using Transformer as a general feature extraction network and used hundreds of times more parameters. ATSFDCNN-CONCAT achieved an accuracy of 86.45% using more than eight hundred times the number of parameters compared with our VaTFER. DSAN-VGG-GENDER and DSAN-RES-GENDER demonstrated an impressive accuracy improvement of 3.83% and 3.05%, respectively, but used several hundred times the number of parameters compared to our VaTFER. In ST-BLNwo/MCD, the accuracy was somewhat higher than VaTFER. However, it came at the cost of having 130 K more parameters than VaTFER. The accuracy of AGCN was much lower than our model, and it used more parameters. Our model enhanced the recognition accuracy by 15.40%, while only utilizing less than half the quantity of parameters compared to L3Net. MicroExpNet and our model had a similar number of parameters. However, the former achieved a lower performance in terms of accuracy. Our model could surpass DDRGCN both in accuracy and model size. As a result, we can conclude that by designing a Transformer-based model in which the relationships among all the AU-based tokens could be effectively modeled, we were usually able to obtain expected results in terms of recognition accuracy and model parameters.

### 4.3. Effect of Different Modules

In this subsection, we evaluate the impact of several important modules involved in the proposed VaTFER model on the performance, i.e., MLCRA, MSA, DLIE, and excitation modules. As shown in Table 4, we amalgamated the various modules mentioned above to create various network models, namely MLSHT, MLMHT, MLDLIET, and VaTFER. MLSHT was only composed of MLCRSA and two simple MLP layers, which is similar to a traditional ViT.

MLSHT was used for evaluating the MLCRSA module. MLMHT used MSA to replace the single-head self-attention mechanism in MLSHT that strengthens the AU-based tokens to acquire complex dependencies among them. MLDLIET consisted of MLCRSA, DLIE, and MLP. VaTFER was composed of MLCRSA, DLIE, and excitation modules. We conducted all experiments on the RaFD dataset. The accuracy of these models with different numbers of MLCRA layers is depicted in Figure 3. Table 5 provides the number of parameters for MLSHT, MLMHT, MLDLIET, and VaTFER with different numbers of MLCRA layers.

From Figure 3 and Table 5, our MLSHT could obtain a 95.05% recognition accuracy with only one MLCRA layer and 26 K parameters. This bested most of the models in Table 1. Furthermore, MLSHT reached the highest accuracy (i.e., 96.11%) when three MLCRA layers were used. As seen, after adding the fourth and fifth layers, the recognition accuracy of the model fiercely decreased. With the increasing number of parameters, such a simple network architecture could not help maintain a relatively smooth accuracy. When focusing on Figure 3, one easily observed that the best accuracy of MLMHT was 96.24%, which was much better than MLSHT. This indicated that replacing single-head self-attention with MSA brought a small increase in terms of accuracy. But using MSA led to a significant increase in number of parameters and computational cost. According to the accuracy shown in green, we found that the module DLIE, which replaced the normal MLP, could help the model MLDLIET reach the best accuracy of 96.4% using only one MLCRA layer. The best accuracy of MLDLIET was higher than that of MLMHT. After adding DLIE, we only used 34 K parameters to achieve the same accuracy as MLMHT with 53 K parameters. The VaTFER model achieved a recognition accuracy of 97.04%, utilizing a combination of DLIE and excitation module to replace two MLP layers. Its accuracy was the best of all the models. Moreover, it only needed 50 K parameters thanks to the introduction of the excitation module, which was far less than the other models. Above all, numerous comparative experiments demonstrated the effectiveness of the different modules. Excitedly, as compared to other models, our VaTFER achieved a better recognition accuracy by combining different modules with fewer model parameters in facial expression recognition tasks.

### 4.4. Netscore of Different Models

To evaluate the rationality of the network structure in the proposed model, we introduced the metric named Netscore, which was proposed in [76]. The Netscore is calculated by $20log(\frac{A{\left(N\right)}^{\alpha }}{P{\left(N\right)}^{\beta }M{\left(N\right)}^{\gamma }})$, where A(*N*) means the network accuracy in percentage, P(*N*) is the network parameters in M-Params (millions of parameters), and M(N) is the number of multiply-accumulate (MAC) operation, which is in G-MACs (billions of MAC operations), and the settings of α, β, and γ were 2, 0.5, 0.5, respectively. We evaluated our model by comparing it with MLSHT, MLMHT, and MLDLIET. We chose the best recognition accuracy in Figure 3 for convenience and comparison fairness. The Netscore values of different models are displayed in Figure 4. The results show that the difference between the scores of our VaTFER and the other three models was small, but the accuracy of our VaTFER was higher.

### 4.5. Comparison of Different Binary Networks

To facilitate the mobile deployment of the proposed lightweight model, we introduced a binarization operation into VaTFER, aiming to reduce the computational complexity and memory access but keeping the least accuracy loss. Here, the binarized VaTFER is termed VaBTFER. In Table 6 and Table 7, we provide the accuracy, computational complexity, and memory access of VaTFER and VaBTFER. Note that the accuracy of VaTFER was previously reported, and we only copied them to Table 6. The computational complexity and memory access are represented using floating-point operations (FLOPs) and member read and write (MemR + W), respectively. FLOPs are obtained from the publicly available package petaflops, while MemR + W is obtained from the torchstat tool package. According to Table 6, VaBTFER can obtain an accuracy comparable to VaTFER. From Table 6, VaBTFER’s and VaTFER’s FLOPs were 13.04 K and 1899.9 K, respectively. Clearly, VaBTFER’s FLOPs dropped by hundreds of times when the binary operation was used. Furthermore, VaBTFER’s MemR + W experienced a decline. To summarize, VaBTFER is more friendly on mobile devices as compared to VaTFER.

In the following, we also compare the simple binarization operation using the sign(·) function with our proposed binarization operation using an extra scaling factor. The results are shown in Table 8. VaTFER-SIGN, DDRGCN-SIGN, and MicroExpNet-SIGN denote the models that apply function sign(·) to directly binarize the weights of VaTFER, DDRGCN, and MicroExpNet. VaBTFER, DDRGCN-SF, and MicroExpNet-SF are the models that adopt the proposed binarization operation with an extra scaling factor. As seen, on the RaFD dataset, the accuracy of VaTFER-SIGN, DDRGCN-SIGN, and MicroExpNet-SIGN decreased by 4.27%, 5.13%, and 4.76%, respectively, as compared with unbinarized VaTFER, DDRGCN, and MicroExpNet. On CK+ and Oulu-CASIA, we achieved a similar conclusion. On the contrary, VaBTFER, DDRGCN-SF, and MicroExpNet-SF had an evident improvement in terms of accuracy when compared with VaTFER-SIGN, DDRGCN-SIGN, and MicroExpNet-SIGN, which showed that adding a scaling factor in the binarization strategy guaranteed achieving a satisfactory accuracy.

## 5. Conclusions

In this paper, we presented VaTFER, a novel lightweight variant Transformer for FER, specifically designed for training on small-scale FER datasets. In comparison to the traditional Transformer encoder, our SCFRT in VaTFER incorporates the MLCRSA module, which efficiently reduces the number of model parameters and models long-range relationships among all AUs. Additionally, we proposed the DLIE mechanism to introduce an inductive bias into the model and make tokens interact. To enhance the model’s ability to learn internal relationships within the class token, we utilized an excitation module before the classification layer. To reduce the computation complexity, we further extended the proposed model to VaBTFER with a binary operation. Through extensive experiments conducted on three FER datasets, we consistently observed that our model achieved satisfactory performance in terms of accuracy and model parameters.

For future research, we intend to focus on two key areas. Firstly, facial expression analysis may encounter more complex challenges, such as lighting, occlusion, and posture, in outdoor scenes. Building a more robust lightweight network is still an urgent problem to solve. Secondly, the performance of using only static features on a video sequence dataset is lower than using spatial–temporal features. Therefore, the fusion of spatial–temporal features and transformers can achieve higher recognition rates.

## Figures and Tables

**Figure 1 sensors-24-00147-f001:**
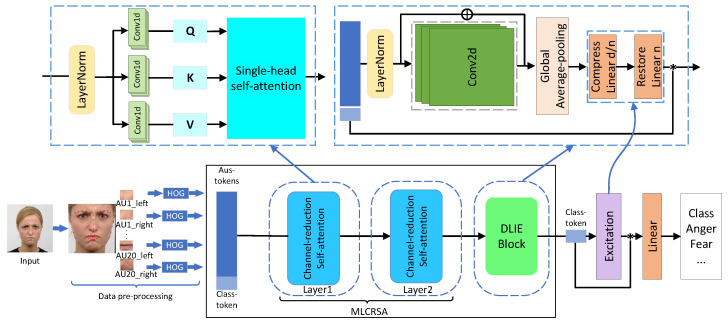
Illustration of our VaTFER: The workflow of VaTFER is shown below and details of one layer of MLCRSA and details of DLIE are shown above. In the workflow of VaTFER, we firstly use AU-based regions after extracting HOG features as AU tokens to replace the simple linear projection patch embedding in ViT. The class token is then added to these AU tokens. The whole tokens are fed into our SCFRT which contains MLCRSA and DLIE modules. The excitation module is used in the final prediction. One layer of MLCRSA contains a layer normalization operation to do the normalization, 1D convolutional layers and single-head self-attention to model long-range dependencies. In DLIE, we first split the AU tokens, operate them through a layer normalization, and then, we use 2D convolutional layers to introduce an inductive bias and fuse them with the original AU tokens. The global average-pooling and two fully connected layers are used to obtain the global information about AU tokens. The global information is multiplied with the class token as the output of our DLIE module.

**Figure 2 sensors-24-00147-f002:**
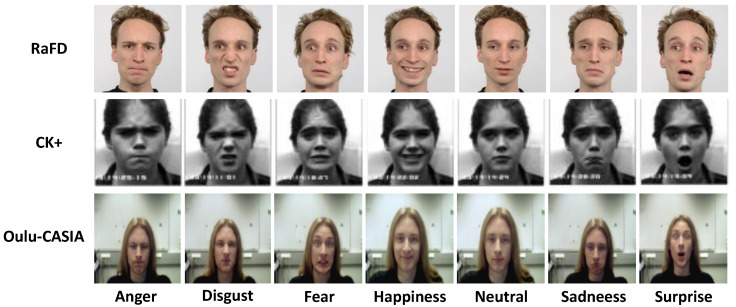
Some samples of RaFD, CK+, and Oulu-CASIA datasets.

**Figure 3 sensors-24-00147-f003:**
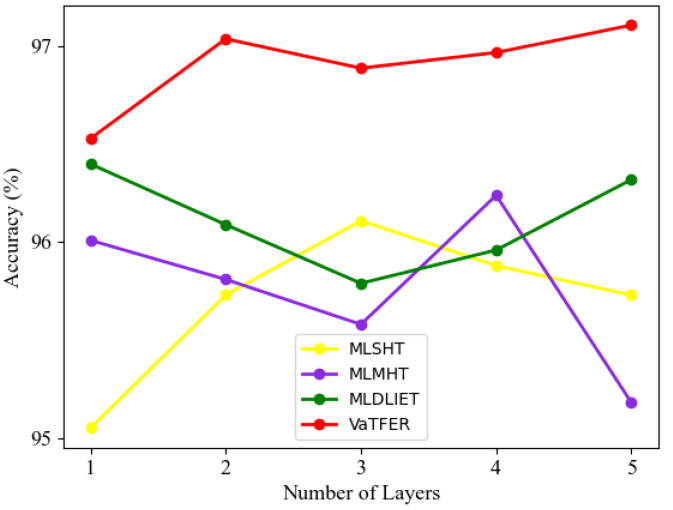
The accuracy of MLSHT, MLMHT, MLDLIET, and VaTFER with different numbers of MLCRA layers on RaFD dataset.

**Figure 4 sensors-24-00147-f004:**
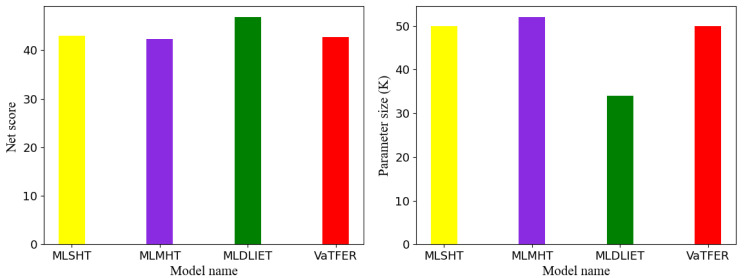
The Netscore of different models for FER.

**Table 1 sensors-24-00147-t001:** Comparisons of different methods on the RaFD dataset.

Method	Accuracy (%)	Parameters (K)
ViT [62]	54.34	86,000
MSC-ViT [62]	98.26	25,360
SqueezExpNet-ABR [63]	96.50	8300
EDNN [64]	95.00	4621
PeleeNet [65]	84.00	2123
ResMoNET [66]	90.00	1721
L3Net [67]	93.25	102
DDRGCN [68]	94.48	110
MicroExpNet [69]	90.08	65
**VaTFER**	97.04	50

**Table 2 sensors-24-00147-t002:** Comparisons of different methods on the CK+ dataset.

Method	Feature	Accuracy (%)	Parameters (K)
Graph + Tran [70]	Static	98.78	24,500
ESTLNet [71]	Static	99.04	20,340
DMSRL-VF [72]	Static	91.26	138,640
ATSFDCNN-CONCAT [38]	Static	98.43	42,000
DSAN-VGG-GENDER [73]	Static	98.60	30,640
DSAN-RES-GENDER [73]	Static	96.98	61,290
ST-BLN wo/MCD [74]	Static	93.19	132.3
AGCN [75]	Static	94.18	143.7
L3Net [67]	Static	94.12	102
DDRGCN [68]	Static	94.32	110
MicroExpNet [69]	Static	90.71	65
**VaTFER**	Static	97.49	50

**Table 3 sensors-24-00147-t003:** Comparisons of different methods on the Oulu-CASIA dataset.

Method	Feature	Accuracy (%)	Parameters (K)
Graph + Tran [70]	Static	89.31	24,500
ESTLNet [71]	Static	89.38	20,340
DMSRL-VF [72]	Static	63.28	138,640
ATSFDCNN-CONCAT [38]	Static	86.45	42,000
DSAN-VGG-GENDER [73]	Static	84.39	30,640
DSAN-RES-GENDER [73]	Static	83.61	61,290
ST-BLN wo/MCD [74]	Static	82.08	132.3
AGCN [75]	Static	75.62	143.7
L3Net [67]	Static	65.16	102
DDRGCN [68]	Static	73.28	110
MicroExpNet [69]	Static	56.71	65
**VaTFER**	Static	80.56	50

**Table 4 sensors-24-00147-t004:** The configuration of different network design based on different modules.

	Modules	MLCRSA	DLIE	MSA	Excitation
Models					
MLSHT		✓	✗	✗	✗
MLMHT		✓	✗	✓	✗
MLDLIET		✓	✓	✗	✗
VaTFER		✓	✓	✗	✓

**Table 5 sensors-24-00147-t005:** The size of the parameters required in MLSHT, MLMHT, MLDLIET, and VaTFER with different numbers of MLCRA layers.

Parameters (K)	Layers	1	2	3	4	5
Models						
MLSHT		26.1	46.3	49.6	52.8	56.0
MLMHT		26.1	46.3	49.6	52.8	56.0
MLDLIET		44.0	54.3	57.5	60.7	63.9
VaTFER		30.4	50.0	53.2	56.4	59.5

**Table 6 sensors-24-00147-t006:** Accuracies of VaTFER and VaBTFER on RaFD, CK+, and Oulu-CASIA.

Datasets	Models	Accuracy (%)
RaFD	**VaTFER**	97.04
	**VaBTFER**	96.39
CK+	**VaTFER**	97.49
	**VaBTFER**	97.03
Oulu-CASIA	**VaTFER**	80.56
	**VaBTFER**	79.38

**Table 7 sensors-24-00147-t007:** The floating-point operations (FLOPs) and member read and write (MemR + W) of VaTFER and VaBTFER.

Models	FLOPs (K)	MemR + W (KB)
VaTFER	1899.9	81.04
VaBTFER	13.04	80.87

**Table 8 sensors-24-00147-t008:** Accuracies of different binary models on RaFD, CK+, and Oulu-CASIA.

Models	RaFD (%)	CK+ (%)	Oulu-CASIA (%)
**VaTFER-SIGN**	92.77	93.04	75.89
**VaBTFER**	96.39	97.03	79.38
DDRGCN-SIGN	89.35	89.73	68.40
DDRGCN-SF	91.85	91.67	71.66
MicroExpNet-SIGN	85.32	85.66	50.34
MicroExpNet-SF	87.66	87.82	52.36

## Data Availability

The data and code used to support the findings of this study are available from the author upon request (leis@njfu.edu.cn).

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
