# Peer review of "VaBTFER: An Effective Variant Binary Transformer for Facial Expression Recognition"

_sensors, 2023, doi:10.3390/s24010147_

Round 1

Reviewer 1 Report

Comments and Suggestions for Authors

This paper proposes an effective and lightweight Variant Transformer for FER called VaTFER. The paper designs an efficient spatial-channel feature relevance transformer module which contains MLCRSA and DLIE modules to handle more parameters and missing inductive bias in the traditional Transformer encoder. Furthermore, the paper uses an Excitation module to replace the vanilla Multi-Layer Perception (MLP) for accurate prediction. To further reduce computation and memory resources, The paper also introduces a binary quantization mechanism, formulating a novel lightweight Transformer model called VaBTFER. However, some parts in the paper would need to be carefully revised.

1. Lines 305-306 of the paper mention that the operation in DLIE is designed to compensate for missing performance utilized single-head self-attention mechanism in MLCRSA. However, why not directly use multi-head self-attention?

2. What is the difference between the FC layer in equation (10) and the Linear layer in equation (3)? Please explain.

3. In Tables 1, how were the experimental data for the ViT, SqueezExpNet-ABR, EDNN and PeleeNet methods obtained?

4. For CK+ and Oulu-CASIA datasets, please add comparisons with Transformer-based methods.

5. MLSHT and MLMHT have the same number of parameters in Table 5. However, it is inconsistent with the description in lines 512-513.

6. The VaTFER method adds the Excitation module over the MLDLIET method. However why does the VaTFER method have fewer parameters in Table 5? Please explain.

Comments on the Quality of English Language

There are some typos in the paper. For example, line 363, 3.4.2 might be 3.4.1. Line 516, “34k” is inconsistent with Table 5. Line 546, Table 6 might be Table 7.

Reviewer 2 Report

Comments and Suggestions for Authors

Authors proposed An Effective Variant Binary Transformer for Facial Expression Recognition. The paper is generally well written and structured but some important points have to be clarified.

a) Provide insights into the specific facial expressions, action units, and spatial-channel features that contribute to the model's predictions. 

b) How Single-head self-attention is better than Multi-head self-attention? Justify the use of Single-head self-attention in your work.

c) Proposed DLIE mechanism in VaTFER be adapted for other vision-related tasks to improve generalization on limited datasets?

d) Consider the recent research to enrich the literature: Multi-Semantic Discriminative Feature Learning for Sign Gesture Recognition Using Hybrid Deep Neural Architecture.

e) How does the performance of VaTFER and VaBTFER translate to user satisfaction and practical applications, especially in emotion-aware systems or human-computer interaction?

f) Ablation study can be included with other transformers.

The paper aims to address the challenges of deploying large transformer-based models for facial expression recognition. It introduces a novel, lightweight VaTFER model and its binary quantized variant, VaBTFER, which incorporates various mechanisms to improve the learning ability, reduce parameters, and enhance prediction accuracy for FER. The experimental results indicate that these models are effective in practice.

Round 2

Reviewer 1 Report

Comments and Suggestions for Authors

I agree to accept this paper.

Comments on the Quality of English Language

None.

Reviewer 2 Report

Comments and Suggestions for Authors

 The authors have addressed all of my queries. I believe the paper can be accepted for publication.